# Predicting Health Risks of Adult Asthmatics Susceptible to Indoor Air Quality Using Improved Logistic and Quantile Regression Models

**DOI:** 10.3390/life12101631

**Published:** 2022-10-18

**Authors:** Wan D. Bae, Shayma Alkobaisi, Matthew Horak, Choon-Sik Park, Sungroul Kim, Joel Davidson

**Affiliations:** 1Department of Computer Science, Seattle University, Seattle, WA 98122, USA; 2College of Information Technology, United Arab Emirates University, Al Ain 15551, United Arab Emirates; 3Lockheed Martin Space Systems, Denver, CO 80221, USA; 4Department of Internal Medicine, Soonchunhyang Bucheon Hospital, Bucheon 420-767, Korea; 5Department of ICT Environmental Health System, Graduate School, Department of Environmental Sciences, Soonchunhyang University, Asan 336-745, Korea

**Keywords:** personalized asthma care, asthma risk prediction, exposome, indoor air quality, logistic regression, quantile regression, transfer learning, sliding window regression

## Abstract

The increasing global patterns for asthma disease and its associated fiscal burden to healthcare systems demand a change to healthcare processes and the way asthma risks are managed. Patient-centered health care systems equipped with advanced sensing technologies can empower patients to participate actively in their health risk control, which results in improving health outcomes. Despite having data analytics gradually emerging in health care, the path to well established and successful data driven health care services exhibit some limitations. Low accuracy of existing predictive models causes misclassification and needs improvement. In addition, lack of guidance and explanation of the reasons of a prediction leads to unsuccessful interventions. This paper proposes a modeling framework for an asthma risk management system in which the contributions are three fold: First, the framework uses a deep learning technique to improve the performance of logistic regression classification models. Second, it implements a variable sliding window method considering spatio-temporal properties of the data, which improves the quality of quantile regression models. Lastly, it provides a guidance on how to use the outcomes of the two predictive models in practice. To promote the application of predictive modeling, we present a use case that illustrates the life cycle of the proposed framework. The performance of our proposed framework was extensively evaluated using real datasets in which results showed improvement in the model classification accuracy, approximately 11.5–18.4% in the improved logistic regression classification model and confirmed low relative errors ranging from 0.018 to 0.160 in quantile regression model.

## 1. Introduction

### 1.1. Asthma and Exposome

Asthma is one of the primary care-sensitive conditions that can be controlled and prevented, through effective care management such as the early recognition of high risk and prompt interventions [1]. A small fraction of asthma exacerbations, which are possibly avoidable by predictive risk modelings account for 63% of the annual total asthma cost in the US and significantly contribute to rising socioeconomic burden [2,3,4,5,6]. The increasing global patterns for asthma and its associated fiscal burden on patients and healthcare providers demand a change to care processes and the way asthma risks are managed. Personalized patient-centered care is a shift from the traditional professional-care model to a service-oriented model aimed at reducing the cost of any symptoms that can be treated outside hospitals. Computing technologies and emerging predictive analysis provide great promise in promoting patient-centered care by empowering patients to participate actively in health risk control [7,8,9,10,11,12].

While factors responsible for increasing the risks of asthma exacerbation are not completely understood, approximately 70–90% of chronic respiratory diseases are attributed to environmental factors, response to which varies significantly over the population [13,14,15]. The term “exposome” refers to the assessment of different environmental exposures’ effect on human health [16,17] and exposures are calculated based on a specific time range, at a specific location and under certain environmental circumstances [18]. Exposome analytics seeks to discover effects of environmental factors on the health of individuals by integrating time and location, as well as behavioral patterns to estimate individual exposure, and then predict health risks of individuals. Indoor air quality, particularly at home, has been recognized as a major source of exposure to hightened asthma triggers [19,20,21]. Most people, elderly people in particular, spend about 80–90% of their time indoors [21,22,23,24]. In addition, the modern home is highly thermally insulated to improve energy efficiency, often to be a detriment to indoor air quality [21]. While the list of known or suspected asthma triggers include many variables (e.g., air pollutants, allergens, certain food, stress, etc.), the present study focuses on indoor air quality that can be monitored on an individual patient-specific basis in real-time and over which patients have significant control in terms of the level of exposure to each.

### 1.2. Asthma Care and Management

Recent health applications appear to positively influence asthma risk management around the globe [25,26,27,28]. Several studies indicate that patients using mobile asthma self-management apps (e.g., AsthmaSense, AsthmaMD, Propelle and ADAM) have significantly improved quality of life scores. Subsequently, those patients were less likely to visit emergency departments due to asthma-related complications [29,30,31]. Additionally, research in [32] proposes a model that incorporates patient’s history of readmission and impacts of patient attribute changes over time on a tree-based classification method to estimate the probability of readmission. In a different direction, several research works present pattern recognition models to find complex interrelations between air pollution, weather, and asthma exacerbation. In [33], authors proposed a method that extracts related features and uses supervised learning approaches such as classification models to detect adverse health events. Overall, however, while systems integrating environment measurement techniques with predictive analytics promise successful implementation of tailoring care to individual patients and thus for transforming the future of healthcare, existing predictive health analytics provide limited help in creating efficient tailored care plans [34]. This continues to be one of the most challenging problems in environmental health research [35,36,37,38].

One major challenge in predictive analytics in asthma is that asthma exacerbations resulting in hospitalization and emergency room visits are rare events and current predictive models exhibit unsatisfactory accuracy for risk analysis of such events [39,40]. This is mainly due to imbalanced datasets where the high risk zone is much smaller than the normal zone but also partially due to the small size of individual patients’ datasets. For example, recording one observation per day provides only 180 data points over a 6 month period and is not sufficient to develop a neural network based model, while several over-sampling methods have been proposed to solve the imbalanced dataset problem [41,42,43,44] in order to improve the accuracy of classification models, little is known about their effectiveness on the models with small sized training data. On the other hand, most machine learning techniques require a large amount of data to train high quality models. This problem is acute for practical and realistic use of predictive models in health applications where datasets for individuals are frequently very small.

An asthma exacerbation can be the result of a single or a mixture of environmental triggers that are of spatiotemporal nature. In addition, measurements of individual exposures in space and time are affected by many factors and governed by complex interactions and relationships between environmental and human systems [45,46]. However, these spatio-temporal properties of environmental data and human behavioral changes are not fully captured in the existing predictive models. The review in [47], which assesses the effectiveness and feasibility of using smartphones and tablet apps to facilitate asthma self care and management, highlights a gap in considering the environmental impact and the seasonal nature of asthma. This environmental impact could improve the efficacy of apps as standalone interventions [48]. Another shortcoming of current predictive models is in interpreting the outcomes of predictions and the lack of guidance that helps health professionals utilize to the largest extent possible their domain knowledge in the process of prediction modeling and applying the results to patient interventions [49].

### 1.3. Our Contributions

In this paper, we focus on the problem of estimating the probability that a patient will experience a critical asthma exacerbation given the patient’s exposures to indoor environmental factors. We address the limitations of existing predictive models and propose a framework to improve the predictions and applicability of logistic regression and quantile regression models. In the framework, two regression models collaborate together to make predictions. As the framework continuously collects data, the models evolve with newly updated parameters and hyper-parameters.

The first improvement applied in our framework is the deep learning technique of transfer learning to overcome the performance issues of logistic regression as classifier due to the limited size of training data. Our proposed TL method trains logistic regression models through three phases, the first phase is training a fully connected neural network source model using population data, and it is further tuned for target model using an individual patient’s data in the second phase. In the last step, the last layer outputs of the target model are inputted to logistic regressors. The second improvement method is a method of variable sliding window to improve the accuracy of individual patient risk estimation regression models.

In classification modeling, we remark that we studied many other classification models for the target model, which are known to have modest data requirements, such as decision trees, random forests, and support vector machines. We found that logistic regression was the best, so for simplicity of exposition, we restrict our attention to that model. Furthermore, while this paper focuses on asthma risk prediction, it is worth mentioning that the proposed solutions can be transferred to other environmental chronic diseases with adjustments.

## 2. Materials and Methods

In this section, we present our approach to design and implement an asthma risk management system. Figure 1 presents the conceptual design model for the proposed framework that consists of the following components: (1) real-time data acquisition and management; (2) exposure estimation and risk prediction; (3) daily interventions and feedback, and assessment. The model integrates measurements of environmental and physiological conditions, estimation of exposure, evaluation of current health state and prediction of adverse health events. In the proposed machine learning framework, individuals’ exposures to indoor environmental factors are collected in real-time along with their daily routines, which are used as parameters to assess the interactions between asthma risk and indoor air quality. The outcomes of the prediction are then used in the targeted interventions to reduce the probability of high risk.

### 2.1. Study Design

#### 2.1.1. Asthma Risk Measurement

The peak expository flow rate (PEFR) measurement is one of the primary health-level indicators used in asthma care and management [50]. One of the uses of this measurement is to quantify an individual’s asthma exacerbation level and provide the medical practitioners with basis for better understanding of the predictions and thus, better decision-making outcomes. The significance of PEFR measurement is catagorized into three zones, green, yellow and red, using a standardized “normal” value that is established by the American Lung Association based on population-level data using gender, age and height information [51]. One drawback of estimating asthma severity based on population-level norms in this way is that the high variability of PEFR values within the population makes its applicability on the individual level unrealistic.

In this study, we base our forecasts on a simplified version of the individual-based asthma risk zoning method proposed in [52]. The method allows for classification of a patient’s condition into several zones based on the patient’s own historical distribution of PEFR values. For the purposes of prediction of high risk days in this paper we use only two zones, a “safe zone” which we nominally take to be PEFR values in the upper 80% of the patient’s historical PERF values and a “risk zone” taken to be PEFR values in the lower 20% of the probability density function distribution. The PEFR value dividing the two zones is called the critical PEFR value, $PEFR_{C}$. The objective of the inference engine will be to predict when a patient is in danger of entering the risk zone, which is understood to be a potential medical emergency where severe airway narrowing is likely to occur and immediate action may be necessary. Therefore, as discussed in [52] it is important for doctors and patients together to analyze the patient’s PEFR distribution as it relates to the patient’s actual health condition and take care to modify the 80/20 cutoff as necessary. Accordingly, the system is able to evaluate the susceptibility of each individual to an asthma exacerbation, on an individual basis as the value of the exposure to variables changes over time.

The models aim to estimate the probability, $P(y<PEFR_{C})$, that a patient’s PEFR value today will fall below their or her critical value. Even though $PEFR_{C}$ may have been established based on the lower, say, 10% or 20% of the patient’s PERF values, the probability of falling below $PEFR_{C}$ depends on many environmental factors and hence use of a various values of $PEFR_{C}$ can be used to measure a patient’s daily health risk. Figure 2 illustrates the distributions of morning PEFR values of the 19 patients who were participants in our study. The blue dotted line represents the average of 20% quantile value of the 19 participants’ PEFR data while the red line inside the boxplot represents 20% quantile value of a particular patient’s PEFR data.

#### 2.1.2. Indoor Air Quality Measurement

Fine particulate matter of 2.5 µm ($PM_{2.5}$) concentration, carbon dioxide ($CO_{2}$), temperature, and relative humidity (dampness) are known to be some of the most important asthma risk factors among the list of known or suspected asthma triggers because they have high temporal variability, are strongly affected by participants’ activities and can be monitored in real-time.

In our risk prediction, we study aggregate exposure to those four variables constructed through various environmental factors and aggregation methods. These four air quality data points were obtained through laser-light scattering sensor with 2 min intervals in participants’ main living spaces. Each participant’s exposures to these variables were estimated using 24 h historical air quality data before the participant’s PEFR measurement. The air quality data stamped by time and location were collected in the individual’s daily routine and stored in our database server for data preprocessing of exposure estimation.

#### 2.1.3. Population Data

A total of 19 participants were recruited from the adult asthma patients (aged 34 to 83 years) who had joined our ESCORT (environmental health smart study with connectivity and remote sensing technologies) study [53]. These participants were consulted and monitored by doctors and medical practitioners at Soonchunhyang University Bucheon Hospital, South Korea. In our study, all participants are non-smokers and occupants at their home are all non-smokers.

The patients’ daily PEFR values were collected twice a day (morning and evening) between 1 November 2017 and 31 May 2018 and the resulting dataset sizes vary between 118 days and 212 days with an average of 154 days. Participants agreed to keeping the air quality monitoring unit for monitoring and storing indoor air quality such as $CO_{2}$, $PM_{2.5}$, temperature and humidity, and writing their daily activities and place visited every 30 min in a diary provided. Several categorical data such as income, living situation and cooking habits were also collectected. The comprehensive nature of our framework offers the opportunity to track spatiotemporal exposure patterns for each participant over a period of time and to capture participants’ daily activities. The environmental variables and measurement in this study are summarized in Table 1.

### 2.2. Exposure Estimation

To simplify the relationship between stochastic, spatio-temporal sequences of pollutant concentration and their physiological consequences, researchers have recently began to entertain the notion of “exposome” [16], while exposome studies have uncovered many important relationships between environment and human health, the assessment of individuals’ exposure to air quality over time has been confined to population averages, rather than individualized estimates. Measurements of individual patient’s exposure to indoor air pollution is affected by many factors, such as concentration of pollution, location/time of the individual, physical activities (exertion) and behavior, and the human system [45,46]. As part of developing a tailored care plan for a patient utilizing indoor air quality control, historical data of a patient’s exposures to the targeted indoor environmental factors needs to be obtained as well as knowledge of the sources of air pollution and underlying characteristics of the exposure [20]. As more sensor technologies become available, they can be used to monitor real-time indoor air quality and to develop analytical models for asthma care management [16,54].

The proposed system calculates the impact of environmental exposure on individuals biomarkers (e.g., lung function level, PEFR) at any given time. It retrieves information on the concentrations of each air pollutant in the air of identified regions and the timeframe over which the exposure occurs. It then uses the general equation for exposure in (1) and more complex integrative models to quantify exposures. Exposure factors refer to any extra information that is required to calculate the exposure amount such as exposure rates or number of possible spaces, activity-patterns and body weight [53].

Each participant’s exposures to environmental variables can be estimated using a time window on historical air quality data before the participant’s PEFR measurement. The time window can be determined by medical expertise, where approximately 24 h window (between yesterday’s AM measurement time and today’s AM measurement time) was used in our experiments. Then the exposure amount for an activity can be calculated as follows:(1)$$E_{j}=\frac{Conc*Inh.Rate}{60\;seconds},$$
where Conc is the environmental concentration (e.g., $PM_{2.5}$) per activity per person measured every 60 s, Inh.Rate is inhalation rate of adult patients based on ages, and *j* is type of activities. In the estimation of daily exposures to $CO_{2}$, temperature and relative humidity, $E_{j}$ is the mean value of the reported values within 24 h time interval.

Equation (Equation 1) indicates the accumulated amount of environmental exposure per minute for a particular activity *j*. Then, the accumulated daily exposure to an environmental variable is calculated by:(2)$$f\left(x_{i},t\right)=\sum_{j=1}^{N}E_{j}T_{i,j},$$
where $x_{i}$ is the space (room) on which the individual stayed, $T_{i,j}$ is the time spent for activity *j* and $E_{j}$ is the exposure amount for activity *j* per minute. The predefined activities and their characteristics can be found in [53].

### 2.3. Risk Prediction Modeling

Our proposed modeling framework utilizes two commonly used machine learning methods in medical applications, (1) logistic regression (LR) together with a neural network based transfer learning (TR) and (2) quantile regression (QR). The LR method, a popular machine learning method for classification problems, estimates the probability of an event (i.e., a binary response), such as positive or negative, based on a given set of independent variables. It is often used in medical domain to predict the risk of developing a given disease, based on observed variables of the patient [55]. In the context of probability modeling, LR finds an optimal $\Theta =\{\theta_{0},\theta_{1},\ldots ,\theta_{n}\}$, the set of coefficients for the linear combination $z=\theta_{0}+\theta_{1}x_{1}+\ldots +\theta_{n}x_{n}$ of the *n* independent variables, which best estimates the probability, *p*, of developing the disease (positive) when substituted into the logistic function $p=g\left(z\right)=1/\left(1+e^{-z}\right)$. On the other hand, in machine learning contexts, LR is often used for binary classification where p≥0.5 results in the model outputting the positive class.

In a traditional clinical use, LR has the advantage of having a probability associated with output metrics that are relatively easily understood. An example of typical outputs would be “Based on an estimation of your exposure to air pollutants in last 24 h, you have a 50% chance of falling below your critical PEFR value today”. The patient can be also guided to follow a medical protocol to prevent an asthma exacerbation. One disadvantage is that patients frequently receive warnings for events that are highly critical but nonethless have low probability. The situation that the critical event does not occur is considered as “false positives”. Therefore, it is important that the system gains more detailed information when LR estimates a non-negligible probability of a critical event. In our proposed two-step system, we use logistic regression only for classification of the patient’s next-day risk state (high-risk or low-risk) and quantile regression to provide more nuanced information to the patient regarding their overall likelihood of experiencing a critial exacerbation event.

The QR method estimates the conditional median or other quantiles of the response variable based on the values of explanatory variables. Linear regression attempts to find the best $\Theta =\{\theta_{0},\theta_{1},\ldots ,\theta_{n}\}$ for the linear equation $y=\theta_{0}+\theta_{1}x_{1}+\ldots +\theta_{n}x_{n}$ of the *n* independent variables, which predicts the average value of the response variable *y*. The average value is referred to as the “conditional mean” of the distribution of *y* given explanatory variables $\left(x_{1},x_{2},\ldots ,x_{n}\right)$. QR on the other hand attempts to model the quantile values of the conditional distribution, hence it can approximate the whole conditional distribution of a response variable *y* [56]. QR has recently found use in many medical applications in which the more extreme values of a patient’s data are of particular interest [57]. In our modeling framework, QR works as the second method.

All machine learning techniques come with a set of advantages and disadvantages: LR models exhibit unsatisfactory accuracy in individual-level health risk prediction application where the size of training datasets is small. Supposing that we collect a patient’s data for 1 year, then the total number of data tuples is 365. Most machine learning algorithms underperform with this small dataset.

On the other hand, one environmental factor may lead to changes in several aspects of the distribution of other environmental variables, including changes in the mean, variability, and severity of extreme cases. Classical quantile regression analyzse a single quantile or several quantiles separately [58]. Thus, performance of QR models in this context can be improved by taking into account time trends for each quantile level and time locality (i.e., recent data is used in training/validation).

We propose solutions to these problems in order to improve the performance of predictive models: (1) LR with neural network based transfer learning, and (2) QR with a variable sliding window method. The two improved methods are presented in the following subsections and the results of the improvement are presented in Section 3.

#### 2.3.1. Logistic Regression Classification with a Neural Network Based Transfer Learning

One of the main challenges to improving prediction quality of LR models is the limited availability of large high quality labeled datasets [59]. The TL technique, one of deep learning techniques in machine learning, can help overcome a scarcity of data by focusing on fine tuning a pretrained model with a small amount of specialized training data [60]. This strategy has shown great promise in the medical field in the context of image analysis of MRI or CT scan data and images [61]. Authors in [62] reported results of a preliminary study of the effectiveness of transfer learning for asthma risk forecasting. Still, however, to date little research has been performed in the context of individual-level health risk prediction with limited training data.

We propose an improved TL + LR classification method as a pipeline: it trains a fully connected neural network (NN) (source model) with population data of the 18 asthma patients (excluding a target patient) and then retrains the NN with a target patient’s data (target model). The output of the last hidden layer of the target model of the NN model is pipelined into a LR model as input data. Finally, the logistic regression produces a prediction decision (classification) with a probability. The process of transfer learning based logistic regression is shown in Figure 3.

#### 2.3.2. Qunatile Regression with a Variable Sliding Window Method

Spatio-temporal analysis using QR has known to be one of the successful machine learning techniques for time-series data prediction in business and economics [63]. Recent work in [58] presents a joint model of QR and temporal variability for finding patterns of climate change by taking into consideration the spatio-temporal properties of the data. Asthma risks are known to be associated with an overall increase or decrease in temperature, humidity and other air pollutants in a specifically defined past time period. Many studies consider air quality and other environmental facotrs in asthma care and management, but the literature is lacking in in-depth analysis of patients’ exposures to these factors in a recent time period.

To improve QR models, we propose a variable sliding window method, where the time window size (the duration of the model construction) and the length of sliding (model usage time) are determined dynamically over time. This method defines two parameters: the window size *W* of the number of data points (i.e., the number of days) for the model training and validation and the sliding size $m^{k}$ of the number of days for the current model usage duration at $k^{th}$ iteration, which is also considered as the time period for the next model development. The dataset within in a given window consists of a training dataset ($D_{train}$) and a validation dataset ($D_{valid}$). Figure 4 illustrates the use of the sliding window method in the QR modeling process.

As an example of the QR modeling, one can build a QR model using the data collected for 45 consecutivie days (W=45), 30 days’ data points ($D_{train}$) are used for training the model and 15 days’ data points ($D_{valid}$) are used for validation of the model. Note that using a larger value for *W* (a larger training/validation dataset) may loose temporal information by the fact that the relationship between health risks and environmental variables is dynamic and typically changes over time. If the quality of the model is acceptable, this model can be used for a certain number of days *m* to predict the health risk of the patient. The values of *m* can be determined based on the outcomes of the validation process. With recent advancements in computing hardware and software, updating the model every day is possible, but for utility and practical use, 7 days (m=7) would be a reasonable value for *m* unless more frequent updates on the model are required to maintain acceptable errors in prediction.

The optimal values of *W* and *m* depend on many application-specific factors including the desired model accuracy, specific nature of the given data and available computing resources, and should be searched during the model development phase. In our study, we analyzed W=35,40,45,50, and m=30,20,10,5,1 to find a good pairing of (*W*, *m*), and the results are presented in Section 3.2.

### 2.4. A Predictive Modeling Framework and Its Use Case

While various statistical methods exist for evaluating the performance of logistic and quantile regression models [57,64], practical interpretation of their metrics is difficult to convey to medical practitioners and patients. Thus, we propose a new predictive modeling framework that yields understandable information on the model’s performance and delivers easy to use the outcomes of predictions.

#### 2.4.1. Training, Validation and Testing

In the QR modeling, training, validation and testing are conducted using a variable sliding window method as described in Section 2.3.2. As *m* number (moving size) of data are collected and augmented to the dataset, the same number of data points that are the least recently colllected are removed from the dataset. On the other hand, the TL + LR modeling uses *k*-fold cross validation, the most commonly used method for classifiers, for model training, validation and testing. Details of the TL + LR modeling process are below.

**Initialization:** For each patient, we collect a set of data consisting of the variables listed in Table 1 and integrate them to a dataset. In the improved LR (NN-based TR + LR) classification modeling, each patient’s integrated dataset is divided to (a) training/validation data according to an 80%/20% split. We also construct (c) a dataset for the source model by combining all patients’ data except the target patient.

**Training and Validation:** For each patient, we build a source model using the entire dataset (c). For overcoming the class imbalance problem, an oversampling technique is used to generate synthetic data using some samples from dataset (a) and these synthetic data are augmented to dataset (a). We then use k-fold cross validation to build a target model using the augmented data, which is split to *k* non-overlappting datasets (called as folds): For *k* rounds of evaluation, k−1 folds are used for training a model and the remaining 1 fold is used for validatin the model. In the training/validation phase, we build *k* models and select the best model by evaluating the models using the standaard evaluation metrics. The metrics we used in out study are presented in Section 3.1.1. Model overfitting and underfitting are also tested using learning curves and training loss. The hyperparameters are selected through extended training and *k*-fold validation processes to avoid over-fitting while to increase the accuracy.

**Testing:** Once the model is trained and validated, the estimation quality of the model is analyzed through the testing phase on the remaining data (b), which are not used for training. The dataset (b) should keep the same data distribution as the patient’s original data. Hence no over-sampling is applied to balance samples among classes. Standard evaluation metrics for classifiers are used to evaluate the model performance. Averages over all test data represent the quality of the models.

#### 2.4.2. Model Use

In use, the patient’s today’s PEFR value and indoor air quality data values in Table 1 are collected, and the patient’s prediction models are used to estimate the patients’ health risk for tomorrow based on the amount exposure to environmental factors and today’s PEFR value. In our proposed framework, the two methods, TL + LR and QR, collaboratively work to make a risk prediction, influence on parameters and hyper-parameters, and evolve in the life cycle of a prediction framework. Figure 5 illustrates the overview of the proposed predictive modeling framework and a use case of the modeling framework. Steps of the use case are:

**Step 1:** The framework starts with the development of a TR + LR model through the training and validating process using the patient’s historical data. In model usage, the model predicts the class of the patient’s next-day health risk state in terms of falling below their $PEFR_{C}$ ($p(PEFR_{C})\geq 0.5$).

**Step 2:** If the model predicts high-risk class, it sends a request to the QR modeling process for prediction for more detailed information. Model parameters and hyper-parameters including $\tau_{C}$ are updated based on the outcomes of the previous step.

**Step 3:** It uses a QR model to predict the PEFR value $PEFR(\tau_{c})$ associated with the critical quantile $tau_{c}$.

Note that the model training and validation process by using the QR method is independent from the TL + LR modeling although they both can provide the information for updating parameters and hyper-parameters.

**Step 4:** If the value of $PEFR(\tau_{c})$ estimated in step 3 falls below its previous value and the drop value is larger than a threshold θ, the system outputs a prediction report. Model parameters and hyper-parameters including $PEFR_{C}$ are updated based on the outcomes of the previous step.

## 3. Results

The experiments on the quality of the predictive modeling were conducted on the 19 patients’ datasets and Table 2 presents a summary of the datasets.

### 3.1. Performance Evaluation of Classification Models

#### 3.1.1. Evaluation Metrics

The confusion matrix is a commonly used method for evaluating clasification models. In a binary confusion matrix, the model performance is evaluated based on the model’s ability to distinguish “positive” data samples from “negative” ones. The confusion matrix is shown in Table 3, where TP represents “True Positive”, the number of positive data samples correctly classified as positive, FN represents “False Negative”, the number of positive data points incorrectly classified as negative, FP represents “False Positive”, the number of negative data points incorrectly classified as positive, and TN represents “True Netagive”, the number of negative data points correctly classified data as negative. The binary confusion matrix can be generalized to the confusion matrix for muti-class classification. For classification problems with multiple classes, one overall quality metric is arrived at by calculating these numbers for each class independently and averaging the results. In our study, of high-risk prediction, our “positive” samples were the data tuples in which a patient’s PEFR value was below the patient’s critical cutoff ($PEFR_{C}$) and the class containing these data is called $Class_{Risk}$ and the class contraining the data above $PEFR_{C}$ is $Class_{noRisk}$.

The following standard metrics that take into account minority classes were used: (1) weighted accuracy = $\frac{TP}{2(TP+FN)}+\frac{TN}{2(TN+FP)}$, (2) sensitivity (also called recall) = $\frac{TP}{TP+FN}$, (3) specificity = $\frac{TN}{FP+FN}$, (4) precision= $\frac{TP}{TP+FP}$, (5) $F_{1}$-score = $\frac{2*precision*recall}{precision+recall}$, and (6) Receiver Operating Characteristic (ROC), while these matrics are equally important for evaluating classifiers, we emphasize the model’s performance on the target class $Class_{risk}$ in the context of risk prediction.

Weighted accuracy is the average of a model’s accuracy rate at classifying positive samples as $Class_{Risk}$ and negative samples as $Class_{noRisk}$. Sensitivity is the model’s success rate at classifying positive samples as positive while speficity is the model’s success rate at classifying negative samples as negative. On the other hand, precision measures what percentage of data tuples that the model classifies as positive are actually positive. Typically, precision decreases as recall increases. $F_{1}$-score is the harmonic mean of sensitivity and precision, which measures the model’s success at both the correct classification of high-risk samples and avoiding the incorrect classification of low-risk samples as high-risk. The area under a ROC curve (denoted as ROC AUC) provides an overall measure of fit of the model. However, ROC AUC does not account for prevalence or different misclassification costs arising from false-negative and false-positive diagnoses [65]. Change in ROC AUC has little direct clinical meaning for medical practitioners. They proposed an alternative analysis based on the change in sensitivity and specificity at clinically relevant thresholds. This analysis provides full benefits of prediction models by incorporating estimates of prevalence and misclassification costs, and hence it is clinically interpretable since it reflects changes in correct and incorrect risk predictions when a new test is introduced.

#### 3.1.2. Classification Model Performance in Risk Prediction

Our model performance improvement focuses on the metic of sensitivity (correctness of the target high risk zone ($C_{p}$ < $PEFR_{C}$) while keeping a good balance in improving all other metrics. Although these model performance metrics assist medical practitioners in integrating them into a care plan, the subtle practical implications of the metrics may be challenging for non machine learning professionals to understand. In fact, metrics such as sensitivity and the $F_{1}$ score are sometimes heavily relied upon the model development and on the training and validation phase rather than on the model usage phase.

For external and internal validation of the models, we divided the dataset to training/validation data (80%) and test data (20%). For each patient’s model, we conducted 10-fold cross validation for source model traning/validation and 5-fold cross validation for target model training/validation. We then evaluated the model on a test data. We describe the models’ performance based on the metrics discused above and present aggregate results and the training loss of the model together with the accuracy.

In our experiments, we first applied the synthetic minority oversampling technique (SMOTE) [41] in the training process of the models to overcome the imbalanced class problem. We then implemented a transfer learning paradigm using deep neural networks and LR models. Table 4 summarizes the results of the stand alone LR models and TL-based LR models. The results show the overall performance gains of the TL-based logistic regression models comparing it to that of the stand alone logistic regression, 14.3% in weighted accuracy, 18.4% in sensitivity, 11.5% in specificity, 13.1% in precision, 15.7% in $F_{1}$ score, and 18.3% in ROC AUC. In the improved LR models with NN-based TL, the average of sensitivity was 0.727 and the average of specificity was 0.757, while those values are 0.614 and 0.679, respectively, in LR models. This shows that the improved LR models provide a more balanced accuracy between positive class $C_{P}$ (<$PEFR_{C}$) and negative class $C_{N}$ ($>=PEFR_{C}$).

Figure 6 illustrates the loss and accuracy of the source model training and validation and those of target model retraining in the NN based TR + LR models. Figure (a) shows training loss and accuracy and figure (c) shows validating loss and accuracy in the training phases for the source model using 24 patients’ datasets (except the target patient SB-078). The loss and accuracy of retraining for the target model for SB-078 are shown in figure (f). Similarly, figures in (b), (d) and (f) show the loss and accuracy of source model training-validation-target model retraining for SB-083.

The TL + LR classification models perform with reasonable accuracy rates for use in health risk prediction as compared to the accuracy of commonly used models in health domains. In Table 4, the average sensitivity of logistic regression models was 63% and this was increased to 70% when the transfer learning technique was applied, which resulted in 11% overall improvement rate. At the same time, the average specificity of the models was also improved from 66% to 74%.

Similar improvement in TL + LR was found in other metrics, such as weighted accuracy, precision, and $F_{1}$ score. The similar improvement trends in model quality can be seen in the results of 19 individual patients’ models as shown in Figure 7. The performance summary of TL + LR for 19 individuals are shown in (a) and (b), respectively. The results also show that TL + LR results in tighter bound in the performance measures.

### 3.2. Performance Evaluation of Quantile Regression Models

To evaluate the quality of the QR model, we used a uniform measure of the relative error for each quantile τ proposed in [66] and evaluated the model through extensive experiments on real patients’ datasets. The uniform measure of the errors is calculated as follows:(3)$$\begin{array}{c} Err_{\tau }=\left|\frac{N_{\tau }}{N}-\tau \right|, \end{array}$$
where $N_{\tau }$ is the number of data points (days having observed PEFR values) under that day’s predicted τ PEFR quantile value and *N* is the total number of test days.

Table 5 shows the mean and standard deviation of the values of the relative error, $Err_{\tau }$, for individuals QR analysis with varying the sizes of training window *W* ($T_{train}$) with 7 days of the model use time *m* ($T_{use}=7$). Our analysis shows that the avarage relative errors of the 19 patients’ models are very low for all τ values ranging from 0.018 to 0.16 on average. A general trend is that small values of τ and large values of τ result in higher errors. Figure 8 shows the relative error in each of 19 patients’ models with 45 days for training window ($T_{train}=45$) and 7 days of the model use time ($T_{use}=7$).

A general trend found is that increasing the sliding window size ($T_{train}$ = 35, 45, 50) reduces the errors but the results also show that increasing the training window size $T_{train}$ does not always reduce the errors for some τ values. We also see that the average $Err_{\tau }$ of the models using 30 day is little lower than that of the model using 45 day window. This means that a model can be developed for a patient in a relatively shorter period (i.e., in 1 month) and the model can be refined further while the system serves the patient risk management. With an optimal window for the QR model for each individual, we show the average relative errors of each individual’s quantile regression analysis for different window sizes in Figure 9.

## 4. Discussion

The ability to control individual asthma attacks caused by environmental triggers contributes to asthma aggravation reduction, and therefore decreases mortality and treatment cost as well. It is well established that machine learning techniques can contribute significantly to the management of asthma exacerbations and the reduction of its risks but the efforts are still minimal. One major challenge in individual-level health risk modeling is that the performance of commonly used machine learning methods is degraded with the small sized training data, which is frequently found in health applications. Moreover, many of these methods often ignore spatio-temporal properties existing in the data.

Another shortcoming is that doctors and patients often can have significant difficulty understanding the outputs from the models and hence arriving at a practical and useful interpretation of the risk prediction with a probability that is associated with the patient’s critical PEFR value $PEFR_{C}$ [49]. Suppose that the system generates a prediction report that the patient’s PEFR value will fall below their $PEFR_{C}$ with a relatively low probability, let us say 20%. A formal meaning of this report is $P(y<PEFR_{C})=0.20$, which can be interpreted to a message that the patient’s falling into the risk zone is unlikely so the patient will struggle only slightly with asthma condition. Does this message deliver a sensitive and useful information that can help the patient?

A different challenge in the use of the probability associated with a health risk is to deal with many false alarms. If the system warns the patient with the probability of 20%, then 80% of the warnings are false warnings. Still most people would like to receive a report when the change of having an asthma exacerbation is 20% to avoid hospital admissions or emergency room visits. Therefore, machine learning techniques to automatically explain the results of risk prediction and provide guidance on how to use the outcomes in an asthma care are critical. This opens the possibility of real-time intervention to minimize asthma risk at home.

In this paper, we propose a modeling framework that incorporates two well-known machine learning algorithms to deliver more accurate predictions hence more effective solutions for progressive, individualized and preventive asthma risk management. As it is one of the major challenges in any individual-level health modeling, the dataset size of each patient is small (mean = 172 days) in our application and needs to be addressed. Training on such a small data set results in relative low accuracy. Our approach is to use a “transfer learning” strategy that incorporates population data as a base model and then refines the model using an individual patient’s data. The results of transfer learning based logistic regression show success of performing transfer learning on all patients’ data for individual’s risk prediction. Our study demonstrates the promise of transfer learning in the development of high quality predictive models based on small dataset.

## Figures and Tables

**Figure 1 life-12-01631-f001:**
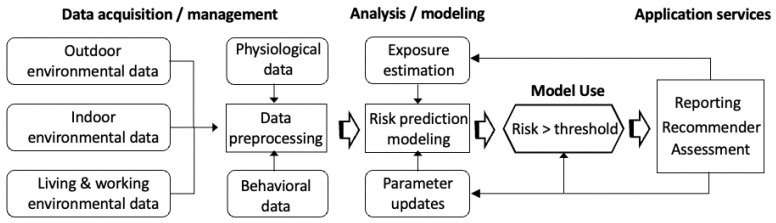
An overview of asthma risk management.

**Figure 2 life-12-01631-f002:**
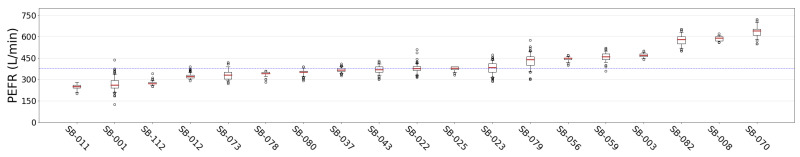
Distributions of participants’ PEFR data.

**Figure 3 life-12-01631-f003:**
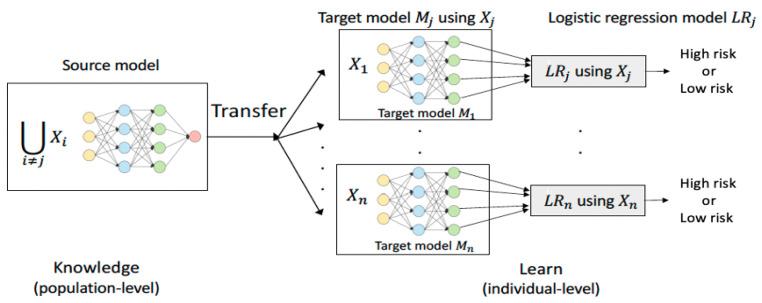
Transfer Learning based Logistic Regression (TL + LR).

**Figure 4 life-12-01631-f004:**
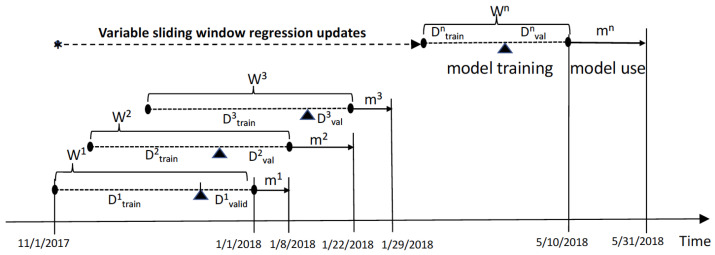
QR with a variable sliding window method.

**Figure 5 life-12-01631-f005:**
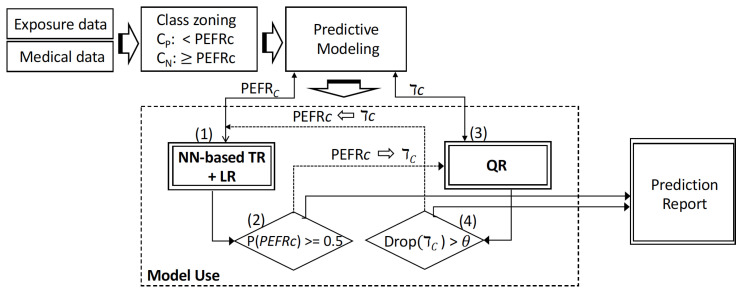
A use case of the predictive modeling framework.

**Figure 6 life-12-01631-f006:**
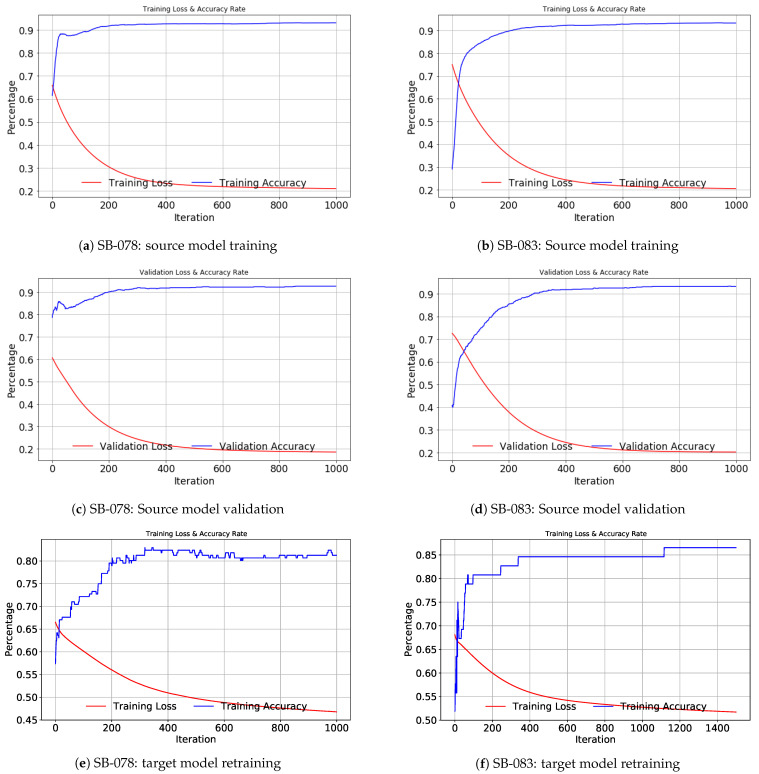
Examples of training loss and accuracy rate of NN-based TL in source model training, validation and target model retraining phases.

**Figure 7 life-12-01631-f007:**
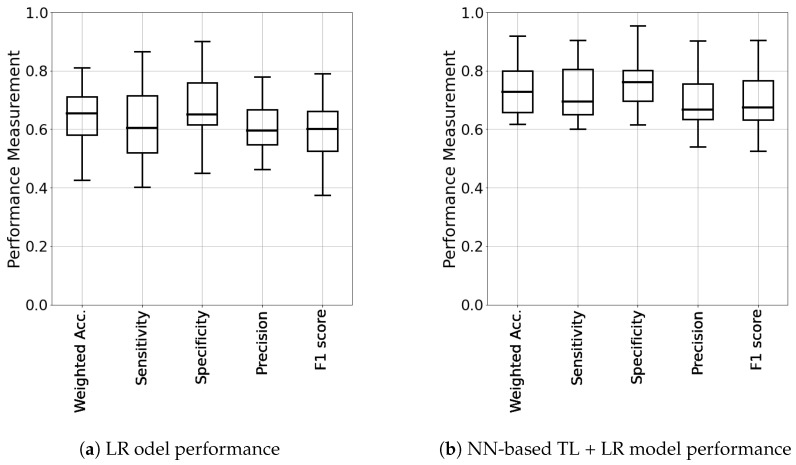
Model performance summary of 19 individuals: LR vs. NN-based TL + LR.

**Figure 8 life-12-01631-f008:**
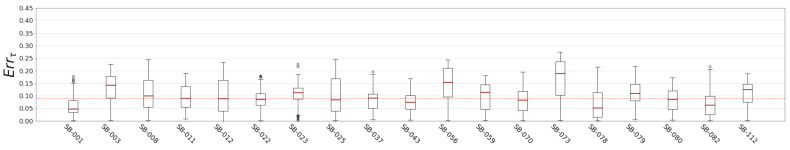
QR relative error analysis of 19 individual models $T_{use}=7$.

**Figure 9 life-12-01631-f009:**
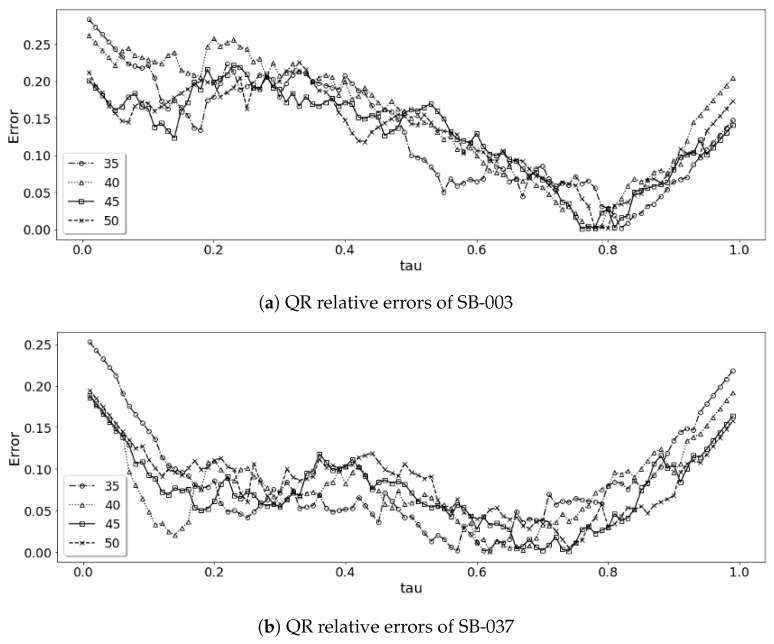
QR relative errors in 4 selected individaul models.

**Table 1 life-12-01631-t001:** Environmental variables and mesurement.

Category	Variables	Measurement
Physiological data	yesterday’s PEFRs	twice a day (AM & PM)
Indoor air pollutants &	*PM*2.5, *CO*_2_	every 60 s interval via
other variables	temperature, humidity	remote sensors installed at home
Cooking behavior	the frequency of frying	level 1 (evey day)–level 7 (none)
Living environment	distance from home to major roads	level 1 (<1 m)–level 5 (>11 m)
Life style	income level	level 1–level 9

**Table 2 life-12-01631-t002:** Data distribution.

		Overall (*n* = 19)			Women (*n* = 10)	Men (*n* = 9)
		P25	P50	P75	P50	P50
Data size	per patient (days)	140	188	196	163	203
Age	(years)	56	72	75	65	68
BMI	(Kg/m^2^)	23.8	21.9	26.8	23.9	23.6
AM PEFR	(L/min)	313.7	373.3	453.9	350.7	462.3
Daily average exposures (24 h)						
	Temperature (°C)	21.9	22.4	23.7	21.2	22.6
	Relative humidity (%)	37.9	32.7	44.1	40.9	37.3
	*PM*_2.5_ (μg/m^3^)	40.2	35.7	50.6	46.2	35.7
	*CO_2_* (ppm)	1005.9	886.9	1241.0	1030.4	918.4

**Table 3 life-12-01631-t003:** Confusion matrix.

	Predictive ${class}_{Risk}$	Predictive ${class}_{noRisk}$
Actual ${class}_{Risk}$	TP	FN
Actual $class_{noRisk}$	FP	TN

**Table 4 life-12-01631-t004:** Average model performance of 19 individuals.

Method	Weighted Accuracy	Sensitivity	Specificity	Precision	$F_{1}$ Score	ROC AUC
LR with SMOTE *	0.645	0.614	0.679	0.607	0.596	0.618
NN-based TL + LR with SMOTE *	0.738	0.727	0.757	0.687	0.689	0.741

* SMOTE: the synthetic minority over-sampling technique [41].

**Table 5 life-12-01631-t005:** Quantile regression analysis with $T_{use}=7$.

*T_train_*		30		35		45		50		Average	
		*Err_τ_* *	std	*Err_τ_*	std	*Err_τ_*	std	*Err_τ_*	std	*Err_τ_*	std
Tau(*τ*)	0.01–0.10	0.092	0.020	0.103	0.016	0.118	0.016	0.037	0.011	0.087	0.016
	0.11–0.20	0.057	0.005	0.112	0.014	0.068	0.018	0.042	0.009	0.070	0.011
	0.21–0.30	0.019	0.020	0.053	0.034	0.021	0.009	0.032	0.006	0.031	0.017
	0.31–0.40	0.030	0.009	0.015	0.010	0.016	0.006	0.012	0.014	0.018	0.010
	0.41–0.50	0.030	0.010	0.009	0.005	0.025	0.015	0.045	0.010	0.027	0.010
	0.51–0.60	0.033	0.015	0.050	0.026	0.048	0.010	0.078	0.021	0.052	0.018
	0.61–0.70	0.046	0.015	0.101	0.006	0.071	0.015	0.055	0.022	0.068	0.014
	0.71–0.80	0.111	0.020	0.105	0.027	0.100	0.012	0.056	0.019	0.093	0.020
	0.81–0.90	0.143	0.020	0.157	0.009	0.152	0.011	0.113	0.023	0.141	0.016
	0.91–0.99	0.173	0.026	0.169	0.023	0.145	0.014	0.153	0.009	0.160	0.018
average		0.0734	0.016	0.0874	0.017	0.0764	0.0126	0.0623	0.0144	0.0747	0.015

* *Err_τ_* = a measure of the error for *τ*, *T_train_* = # of days of model training, *T_use_* = # of days of model use.

## Data Availability

Data cannot be shared publicly because of personal data protectionguideline. Data are available from the Soonchunhyang Risk Assessment Center of DataAccess/Ethics Committee (contact via leesr@sch.ac.kr or phil.cjs@gmail.com) for researchers who meet the criteria for access to confidential data.

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
