# Peer review of "Predicting Health Risks of Adult Asthmatics Susceptible to Indoor Air Quality Using Improved Logistic and Quantile Regression Models"

_life, 2022, doi:10.3390/life12101631_

Round 1

Reviewer 1 Report

This manuscript is among the most clearly written ones I’ve reviewed in the past few years. As an air quality specialist, I am not qualified to comment on the machine learning algorithms used to estimate population and individual exposure risks. But the predictive modeling framework makes full sense to me. I would like to see more applications of the same or similar framework to exposure assessment, given the increasing deployment of wearable monitoring devices.

Author Response

The attached file includes authors' responses to reviewer's comments. 

Reviewer 2 Report

In the paper “Predicting health risks of adult asthmatics susceptible to

indoor air quality using improved logistic and quantile

regression models”, the authors present a new approach to predict the risks of patients with asthma using a new model. The paper is novel, interesting and battles an important aspect in epidemiology. Moreover, the paper is clearly and well-written. However, there are certain issues that should be addressed

When building a new statistical predictive model to determine-predict health risks, this model should undergo specific tests, using statistical tests. Specifically, in every new model the authors should

-compare the data fitting of the new versus the old model, we (using−2 log likelihood statistics)

-calibration of the model (Hosmer-Lemeshow test)

-discrimination ability (using receiver operating characteristic (ROC) curve)

-internal validation

-external validation

Author Response

The attached file includes authors' responses to the reviewer's comments. 

Round 2

Reviewer 2 Report

the authors covered all my concerns